# OpenReview forum: "But what is your honest answer? Aiding LLM-judges with honest alternatives using steering vectors"
_ICLR.cc/2026/Conference — Submitted to ICLR 2026_

### Official Review · Reviewer_To6N · 2025-10-30

**Soundness:** 3
**Presentation:** 3
**Contribution:** 3
**Rating:** 4
**Confidence:** 4

**Summary:**

The paper introduces JUSSA, a framework that uses steering vectors to help large language model judges better detect subtle dishonesty such as sycophancy and manipulation by generating more honest contrastive alternatives. Experiments on new manipulation and sycophancy datasets show that JUSSA improves dishonesty detection for weaker judges on simple tasks and stronger judges on complex ones by increasing the score difference between honest and dishonest responses.

Overall, this work presents a creative and promising idea: using steering not to align but to evaluate models. The approach is simple and conceptually distinct from existing methods. However, the experimental scope and comparative analysis are limited, making it difficult to judge the method’s relative effectiveness or scalability. A stronger positioning within the literature and additional baselines would considerably improve the paper’s impact.

**Strengths:**

- The paper addresses an important and under-explored problem: detecting dishonesty in LLM behavior and improving judge reliability. This direction is timely and impactful for the trustworthy evaluation of AI systems.
- The core idea (repurposing activation steering as a means to provide contrastive auditing) is novel and conceptually strong. It reframes steering from alignment enforcement to judgment assistance, allowing evaluators to expose hidden model biases through counterfactual honesty generation.
- The method is lightweight and efficient, requiring minimal data and computational resources compared to prior large-scale steering approaches. This makes it accessible and practical across different model families.
- The paper introduces a carefully curated dataset for nuanced dishonesty detection. The manual curation and validation steps demonstrate attention to quality and ensure realistic coverage of manipulative behaviors that are difficult for LLMs to detect.

**Weaknesses:**

- **Limited Related Work Coverage.** While the paper provides a focused and relevant related work section discussing dishonesty benchmarks and the foundational steering vector methods underpinning the proposed approach, it falls short of adequately situating its contributions within the broader landscape of techniques aimed explicitly at improving judge LLM capabilities. Instead, it primarily focuses on methods closely aligned with its own framework. I would encourage to expand their related work discussion to a broader and more systematic review of recent research on LLM-based evaluation and dishonesty detection would better contextualize the contribution and demonstrate deeper engagement with parallel efforts in this space.

- **Missing Baseline Comparisons.** The paper does not provide direct quantitative comparisons to recent judge-improvement frameworks (e.g., [1][2]. Without these, it is difficult to assess whether JUSSA offers state-of-the-art performance or mainly a conceptual novelty.

- **Limited Model Scale.** Experiments are restricted to Gemma-2-2B. While this choice demonstrates feasibility, it does not fully capture challenges that arise when scaling steering techniques to larger, more complex models. Validation on higher-capacity models (e.g., Gemma-4B or Llama-8B) would strengthen claims of generality and robustness.

- **Benchmark Generalization.** The evaluation relies primarily on the authors’ own datasets. Applying JUSSA to existing honesty benchmarks (e.g., MASK [3], BeHonest [4]) would allow fairer and more interpretable comparisons against prior work.

  - [1] Zhang, Qiyuan, et al. "Crowd comparative reasoning: Unlocking comprehensive evaluations for llm-as-a-judge." arXiv preprint arXiv:2502.12501 (2025).‏
  - [2] Fujinuma, Yoshinari. "Contrastive Decoding Mitigates Score Range Bias in LLM-as-a-Judge." arXiv preprint arXiv:2510.18196 (2025).‏
  - [3] Ren, Richard, et al. "The mask benchmark: Disentangling honesty from accuracy in ai systems." arXiv preprint arXiv:2503.03750 (2025).‏
  - [4] Chern, Steffi, et al. "Behonest: Benchmarking honesty in large language models." arXiv preprint arXiv:2406.13261 (2024).‏

**Questions:**

1. Why weren't other judge-improvement baselines included in the evaluation?
2. How does JUSSA’s performance and efficiency (practicality) compare to other judge improvement methods?
3. Do you anticipate challenges when applying steering-based auditing to larger architectures (e.g., Llama-8B)? If so, what modifications might be required?

---

> ### Author Response · Authors · 2025-11-24
>
> We thank the reviewer for recognizing that our work addresses an important problem, noting our careful dataset curation. We're pleased you found the core idea novel and conceptually strong.
>
> We have made substantial revisions based on your feedback and that of other reviewers. We address each concern below:
>
> > **W1: "Limited Related Work Coverage"**
>
> We agree and have expanded our related work section to better situate JUSSA within the LLM-as-a-judge literature. We've added discussion of pairwise comparison approaches and recent judge-improvement strategies, including the references you suggested. We now more clearly explain how JUSSA differs from existing pairwise methods, mentioning that while those methods compare responses from different models or prompts, JUSSA uses steering to reveal a specific model's internal representation of honesty, which is particularly valuable for detecting when that model is being dishonest relative to its own beliefs.
>
> > **W2: "Missing Baseline Comparisons to judge-improvement frameworks"**
>
> We want to clarify our contribution and scope. Our primary goal is to introduce and validate steering-based evaluation as a new approach for dishonesty detection, rather than to achieve state-of-the-art performance across all judge-improvement scenarios.
>
> To address whether JUSSA's benefits come from steering specifically or just from providing any alternative, we've added two baselines in Section 6.3.1. Temperature resampling (T=1.0) achieves AUROC of 0.778 (worse than single-response baseline of 0.846), while honesty prompting achieves 0.876. JUSSA with steering achieves 0.911, confirming that steering itself adds meaningful value.
>
> Regarding the suggested references, we now cite Zhang et al. [1] in our related work to help frame our work better, see our response to **Q1** for more information.
>
> > **W3. “Limited Model Scale. Experiments are restricted to Gemma-2-2B.“**
>
> Based on your feedback, we have now run experiments with Llama-3.1-8B-it. Full results are in the updated paper. In short, we find similar patterns to Gemma-2-2b-it, with global AUROC scores of 0.832 (Single), 0.903 (Paired), and 0.892 (JUSSA). This suggests that steering-based auditing scales to larger architectures without major modifications, though we expect that architecture-specific hyperparameter tuning (e.g., optimal layer selection) may further improve results.
>
> > **W4: "Benchmark Generalization:” apply to existing honesty benchmarks (MASK, BeHonest)**
>
> We appreciate these references and already mention MASK in our related work. However, we want to clarify an important distinction. MASK and BeHonest measure whether models themselves behave honestly (they benchmark model honesty), while our datasets measure whether judges can detect subtle dishonesty in model outputs (we benchmark judge capability). These serve fundamentally different purposes and don't provide a straightforward way to evaluate whether JUSSA improves dishonesty detection. That said, exploring how JUSSA might be adapted to those benchmarks would be interesting future work.
>
> **We now address the remaining questions**
>
> > **Q1: “Why weren't other judge-improvement baselines included in the evaluation?”**
>
> Our primary contribution is demonstrating the viability of steering-based evaluation as a new approach. We've added baselines (resampling and prompting) to isolate steering's contribution. A comprehensive comparison with methods like [1] and [2] would require substantial additional work and is outside our current scope, though we encourage future work in this direction.
>
> > **Q2: “How does JUSSA’s performance and efficiency (practicality) compare to other judge improvement methods?”**
>
> JUSSA is lightweight by design as it requires only a single example to train the steering vector and adds minimal computational overhead (one additional forward pass). This contrasts with methods requiring judge fine-tuning, multiple crowd responses, or complex decoding procedures. While we haven't conducted direct efficiency comparisons, JUSSA's simplicity is a practical advantage for deployment.
>
> > **Q3: “Do you anticipate challenges when applying steering-based auditing to larger architectures (e.g., Llama-8B)? ”**
>
> Our new Llama-3.1-8B-Instruct experiments suggest JUSSA generalizes well to larger models with similar performance patterns. The main considerations are selecting an appropriate layer (our layerwise experiments indicate there's a suitable range) and choosing an effective steering sample. We found that we needed a different sample for Llama than for Gemma, as Llama refused to answer the original question. This revealed two requirements for effective steering: the model should respond more dishonestly under provocation than at baseline, and it should still answer the base prompt (otherwise steering can cause mode collapse). These properties hold for most responses from both Gemma and Llama.

---

### Official Review · Reviewer_8vA5 · 2025-10-31

**Soundness:** 3
**Presentation:** 4
**Contribution:** 2
**Rating:** 4
**Confidence:** 4

**Summary:**

This paper introduces Jussa (Judge Using Safety-Steered Alternatives), a framework that uses steering vectors to generate honest alternative responses, helping LLM judges detect subtle forms of dishonesty. The paper also introduces a new manipulation dataset covering different categories across three manipulation types.

**Strengths:**

- using steering vectors to help judges instead of fixing models is novel to the best of my knowledge and also interesting
- the method is clear and the empirical results are promising

**Weaknesses:**

- limited models tested: for the LLM-agent it was tested a single model (gemma-2b) making this study limited because you cannot generalize the finding without looking if other models may exhibit different patterns
- the steering vector is trained on only one example from a single category. This poses a limitation as it's not understood how the sample was chosen, if the results are sensitive on choosing this sample etc
- the dataset being synthetically generated raises concerns on its quality
- missing comparison with several related works such as: [1], [2]
- (minor) figure 1 is blurred and this should be fixed

[1] Gu, Jiawei, et al. "A survey on llm-as-a-judge." arXiv preprint arXiv:2411.15594 (2024).
[2] Zheng, Lianmin, et al. "Judging llm-as-a-judge with mt-bench and chatbot arena." Advances in neural information processing systems 36 (2023): 46595-46623.

**Questions:**

- does the findings are similar if another LLM-agent is employed? (maybe larger ones)
- ablation studies on how to choose the samples to optimize the steering vector.

---

> ### Author Response · Authors · 2025-11-24
>
> We thank the reviewer for recognizing the novelty of using steering vectors to aid judges rather than fix model behavior, and for finding our method clear with promising empirical results.
>
> We have made substantial revisions based on your feedback and that of other reviewers. We address each concern below:
>
> > **W1: “Limited models tested: for the LLM-agent it was tested a single model (gemma-2b)** making this study limited because you cannot generalize the finding without looking if other models may exhibit different patterns”
>
> We agree this was a significant limitation. We have now run experiments with Llama-3.1-8B-Instruct, with full results in Section 6.3.3 and Appendix B. The results show similar patterns to Gemma-2b: Single (0.832),  Paired (0.903), JUSSA (0.892). This suggests the approach generalizes across different model architectures, though we note that optimal hyperparameters may vary by model (we did not extensively tune for Llama due to computational constraints).
>
> > **W2: “The steering vector is trained on only one example from a single category.** This poses a limitation as it's not understood how the sample was chosen, if the results are sensitive on choosing this sample etc”
>
> We agree and appreciate the reviewer for pointing out this weakness. We have added an ablation study where we selected an arbitrary sample (the first one in the dataset) from each of the 9 verified categories and re-ran our experiments to assess sensitivity to sample choice. The results (see Section 6.3.2) show small variance across samples, with JUSSA outperforming the Single judge setup in 8 out of 9 categories. Several categories show improvements similar to our main results, which strengthens the robustness of our approach. We appreciate this suggestion as it helps clarify the robustness of the single-shot steering method for our use-case.
>
> Interestingly, we found that we needed to use a different sample for steering with Llama than with Gemma, as Llama refused to answer the original user question for the base response. This revealed two requirements for effective honesty steering: (1) the model should respond more dishonestly to the provoked prompt than to the base prompt, and (2) the model should still provide an answer to the base prompt, otherwise steering can cause mode collapse where the model refuses all queries. However, we note that these properties hold for most responses from both Gemma and Llama.
>
> > **W3: “The dataset being synthetically generated raises concerns on its quality”**
>
> We acknowledge this limitation. Since we investigate dishonesty in machine-generated responses, some degree of synthetic construction is inherent and difficult to avoid while maintaining experimental control. However, we conducted extensive validation of our dataset (Section 5 and Appendix A), including human annotation studies (130 examples across 13 categories), manual inspection of response patterns, correctness evaluations, and embedding similarity analysis. Human annotators confirmed that provoked responses were more manipulative than base responses and that steering reduced manipulation, which corroborates the LLM judge assessments. We believe this extensive validation provides reasonable confidence in the dataset quality, and we think the dataset will be valuable for others studying manipulation detection.
>
> > **W4: “Missing comparison with several related works such as: [1], [2]”**
>
> Thank you for these references. We have expanded our related work section to better position our contributions within the LLM-as-a-judge literature, including discussion of these works and how JUSSA differs from standard judging approaches.
>
> For **W5**: We have updated the figure now. Thanks for pointing this out.
>
> **Regarding the questions:**
> We believe our additions, particularly the Llama experiments and sample selection ablation, directly address your remaining questions about generalizability and robustness, but please let us know if we missed anything there.

---

### Official Review · Reviewer_85f2 · 2025-11-01

**Soundness:** 2
**Presentation:** 3
**Contribution:** 2
**Rating:** 2
**Confidence:** 4

**Summary:**

This paper proposes JUSSA, a framework that doesn’t steer the task model to be safer for its own sake, but uses a single sample steering vector to generate "more honest" alternative responses that are then shown to an LLM judge for contrastive evaluation. The key hypothesis is that judges struggle to spot subtle dishonesty in isolation, but become more accurate when they can compare the original response against a steered "honest" alternative produced from the same agent. The steering vector is trained by promotion steering to increase the likelihood of the base (honest) response when the model is prompted with the provocation; at inference time it is applied to layer 10 (residual stream) of Gemma‑2b‑it to produce steered variants for both base and provoked prompts. Detection is scored via AUROC over judge assigned "manipulation/sycophancy" scores for base vs. provoked responses.


To test the idea, the authors adapt a sycophancy benchmark and introduce a model written manipulation dataset spanning 13 categories (later narrowed to 9 after QC). A paired judge sanity check and a 130‑example human study assert that provoked answers are more manipulative and that steering reduces manipulation while improving correctness. Main results: for manipulation detection, JUSSA improves AUROC for GPT‑4.1‑base judges from 0.893 -> 0.946 and Claude‑3.5‑Haiku from 0.859 -> 0.929. For GPT‑4.1 variants, JUSSA hurts nano on manipulation (0.756 -> 0.677) but helps base (0.777 -> 0.870), for sycophancy it helps small (nano 0.709 -> 0.848) yet hurts large (base 0.957 -> 0.907) judges. A layer wise analysis suggests steering is most effective in mid layers (~8-13) where representations of honest vs. provoked trajectories diverge.

**Strengths:**

- Novel framing: Repurposes steering vectors for judging rather than controlling model behavior. This reframing of steering as an evaluation aid is conceptually creative and aligns with recent interest in interpretability-aided evaluation.

- Layer analysis: Provides a careful exploration of where steering is most effective, showing that layers 8–13 are most sensitive to honesty steering, consistent with known interpretability findings about mid-layer semantic representations.

- Empirical signal: Demonstrates consistent improvements in AUROC on manipulation detection across most categories for major models like GPT-4.1-base and Claude Haiku, suggesting practical potential in enhancing judge discrimination.

- Lightweight setup: Uses a single sample "promotion steering" approach to train an honesty vector efficiently, avoiding large labeled datasets and making replication feasible.

- Human validation: Includes human evaluation confirming that provoked responses are more manipulative and that steering mitigates manipulation, lending some empirical support to the central hypothesis.

**Weaknesses:**

Prompt leakage confound: In the judge prompt, the “steered” output is explicitly labeled as the “more honest alternative,” which likely biases the judge toward assigning it higher honesty scores. This framing undermines the central claim that JUSSA’s improvements arise from genuine model-level contrast rather than prompt wording.

- Lack of proper baselines: The paper does not compare JUSSA against simple black box contrastive setups, such as generating a second "objective" or "selfcritical" response without steering. Without these, it is unclear whether steering contributes anything beyond giving the judge a comparison.

- Limited generalization: The approach is tested only on one agent model (Gemma-2b-it). There are no results demonstrating cross model transfer or replication on other architectures like Llama or Mistral, limiting external validity.

- Dataset construction issues: The manipulation dataset is model written and partially filtered after evaluation, with four categories dropped due to poor quality. This post-hoc pruning weakens claims about robustness and can inflate reported performance.

- Missing statistical rigor: The results report raw AUROC values but omit confidence intervals, significance testing, or multi seed variation. Moreover, the sycophancy results show performance drops for stronger judges, yet this regression is not deeply analyzed.

**Questions:**

Q1. Judge prompt confound: In the paper the judge prompt explicitly refers to “Response B – more honest alternative response.” Can you confirm whether this phrasing was included verbatim in your main experiments? If so, have you re-run the evaluation with neutral wording (e.g., “Response A” and “Response B”) to quantify how much of the AUROC gain remains once this label bias is removed?

Q2. Contrastive baseline comparison: Since JUSSA provides a second, steered response for contrast, how much of the improvement comes simply from having any contrastive pair versus specifically from the steering vector? A strong rebuttal could include results for simple non steered alternatives (e.g., temperature sampling, "be objective" prompting, self critique generation) to show that the steering signal itself adds value.

Q3. Cross model and reproducibility concerns: All experiments use Gemma-2b-it as the agent. Could you show at least one replication on a different base model (e.g., Llama 2/3 or Mistral or Qwen) to demonstrate that the single sample honesty vector generalizes? If not, can you discuss how architecture specific factors (layer size, activation norms, residual design) might affect steering transferability?

---

> ### Author Response · Authors · 2025-11-24
>
> We thank the reviewer for the detailed feedback and for recognizing the novelty of our work, the careful layer analysis, and our human validation efforts.
>
> We have made substantial revisions based on your concerns and those of other reviewers. We address each weakness below:
>
> > **W1: "Prompt leakage confound: […] 'steered' output is explicitly labeled as 'more honest alternative,' which likely biases the judge toward assigning it higher honesty scores"**
>
> We want to clarify how our evaluation works, as we believe there may be a misunderstanding. We never compare a base response directly against its steered version in the same prompt. Instead, base responses are always labeled "Response A" (paired with steered-base as "Response B"), while provoked responses are also always "Response A" (paired with steered-provoked as "Response B"). Our AUROC metric measures how well judges distinguish base from provoked responses.
>
> Since both conditions see the same label ("Response A"), the phrase "more honest alternative" for Response B affects both equally and **cannot unfairly bias the comparison between base and provoked responses in a way that favors the JUSSA over Single Judge**. The phrase may shift absolute scores upward, but it doesn't change the relative difference that determines AUROC. In early experiments we found that removing the phrase produced minor reduction in AUROC. We kept it because it helps judges understand that Response B represents the model's own honesty direction, so if response B would have false information, JUSSA would likely not work well, highlighting that a good honesty steered alternative is necessary for JUSSA to work.
>
> > **W2: "Lack of proper baselines: The paper does not compare JUSSA against simple black box contrastive setups"**
>
> We agree and have added two baselines in Section 6.3.1. temperature resampling (T=1.0) and honesty prompting ("Please be objective and honest"). Temperature resampling achieved AUROC of 0.778 (worse than single-response baseline of 0.846), while honesty prompting achieved 0.876. JUSSA with steering achieved 0.911, confirming that the steering vector itself adds meaningful value beyond simply providing a contrastive pair.
>
> > **W3: "Limited generalization: The approach is tested only on one agent model (Gemma-2b-it"**
>
> We have added experiments with Llama-3.1-8B-Instruct in Section 6.3.3, with full results in Appendix B. The results show similar patterns to Gemma, with Single at 0.832, Paired at 0.903, and JUSSA at 0.892. While gains are slightly smaller for Llama, we did not perform hyperparameter optimization due to computational constraints, so better steering configurations could likely improve results.
>
> > **W4: "Dataset construction: […] This post-hoc pruning weakens claims about robustness and can inflate reported performance."**
>
> We filtered categories based on human evaluation quality, not JUSSA performance. Human annotators could not reliably distinguish manipulation levels in these 4 categories, suggesting the provocation didn't work well. Both Single judge and JUSSA are evaluated on the same filtered set of 9 categories, so filtering doesn't favor JUSSA. Our evaluation metrics in Appendix A show the remaining categories have consistent quality signals from both humans and LLM judges. The filtering does show limitations of the way we generate the dataset, yet due lack of existing alternatives of obtaining datasets with subtle differences in dishonesty of model responses, we wanted to be transparant about our process.
>
> > **W5: "Missing statistical rigor: […] omit confidence intervals, significance testing, or multi seed variation. Moreover, the sycophancy results show performance drops for stronger judges, yet this regression is not deeply analyzed"**
>
> We acknowledge that confidence intervals would strengthen our results, however, we used temperature 0 for all models to ensure reproducible outputs. With deterministic sampling, running multiple seeds would produce identical outputs, making traditional confidence intervals less meaningful.
>
> Regarding the sycophancy performance drops, we expect that the strong judges already near ceiling performance (GPT-4.1-base at 0.957) have little room for improvement, and steering can introduce slight noise. Our analysis (Appendix A.4, Table 5) shows sycophancy responses have less variation (MECS: 0.978-0.987) than manipulation responses (MECS: 0.954-0.962), which may explain why steering has less consistent benefits there.
>
> **Regarding the Questions:**
> We aimed to have addressed your questions in our responses above. W1 explains the prompt design, W2 demonstrates steering adds value beyond simple contrast, and W3 shows generalization to Llama-3.1-8B-Instruct.

---

### Official Review · Reviewer_nxs6 · 2025-11-06

**Soundness:** 3
**Presentation:** 3
**Contribution:** 3
**Rating:** 6
**Confidence:** 3

**Summary:**

This paper proposes JUSSA (Judge Using Safety-Steered Alternatives): instead of steering a task model to behave honestly at deployment, it uses a lightweight steering vector during inference to produce a more-honest counterfactual answer from the same model, and gives the judge a pair (original vs. steered) to evaluate. The hypothesis is that contrast along an honesty axis helps LLM-judges detect subtle dishonesty (sycophancy/manipulation) better than single-response judging.

The intuition is clear - it is helpful to have a frame of reference for what "good" looks like. I imagine this is especially true for LLM judges.

The steering vector is learned from a single example via promotion steering on a middle layer’s residual stream, optimizing the model (under a provocation prompt) to prefer tokens of the honest/base response. Experiments use Gemma-2B-it as the agent and multiple judges (GPT-4.1 nano/mini/base; Claude-3.5 Haiku) on (i) an adapted sycophancy set and (ii) a new manipulation dataset (13 categories → 9 after quality filtering). AUROC improves with JUSSA for most manipulation categories with capable judges; small judges benefit notably on sycophancy, but JUSSA can mildly hurt when judges already perform near-optimally or when the task exceeds capacity. A layer study shows middle layers (~8–13) are most effective, aligning with measured representation divergence between honest and provoked trajectories.

Contributions: (1) a practical contrastive auditing recipe using steering for evaluation; (2) a manipulation dataset with human validation; (3) analysis of where honesty steering works best.

**Strengths:**

- Simple, intuitive, and original idea: Reframes steering as an evaluation aid. Contrastive, honesty-steered alternatives give judges a cleaner signal than single-response scoring. As far as I'm aware, I have not seen steering used for this purpose.

- Lightweight steering: Single-sample promotion steering suffices to generate informative counterfactuals.

- Layer insight: Middle-layer effectiveness matches representation divergence analysis; actionable for activation-space methods.

- Careful validation: Human annotators corroborate that provoked > base in manipulation, and steered-provoked reduces manipulation; four weak categories are transparently filtered out.

- Candid limitations: Authors note where JUSSA can hurt (easy tasks with strong judges; hard tasks with weak judges).

**Weaknesses:**

- Missing baselines: No comparison to (a) k-sample self-contrast (two unsteered samples), (b) prompt-only “honesty” contrast (no steering), or (c) SAE-targeted/CAA steering as alternative ways to generate contrasts.
- Limited models: Agent is only Gemma-2B-it; unclear robustness across agent sizes/architectures or different provocation styles.
- Residual stylistic cues: Although disclaimers are partially analyzed, gains may still be partly driven by surface cues (length, boilerplate)? Stronger cue-controls would help.
- Dataset external validity: Manipulation set is largely model-authored and curated; broader inclusion of human-written prompts would make it more realistic.

**Questions:**

If the authors could address any of the weaknesses above, particularly around the baselines, I'd be open to change my reviews.

---

> ### Author Response · Authors · 2025-11-24
>
> We thank the reviewer for the thorough and constructive review, and for recognizing our contributions.
>
> We have made significant additions to the paper based on your feedback and that of other reviewers.
>
> > **W1: "Missing baselines: No comparison to (a) k-sample self-contrast (two unsteered samples), (b) prompt-only “honesty” contrast (no steering), or (c) SAE-targeted/CAA steering as alternative ways to generate contrasts."**
>
> We have added two key baselines in Section 6.3.1. Resampling with temperature 1.0 achieves an AUROC of 0.778, which actually degrades performance below the single-response baseline (0.846). Honesty prompting ("Please be objective and honest in your response") performs better at 0.876 but still falls short of JUSSA with steering vectors (0.911).
>
> These results show that steering itself adds meaningful value beyond simply providing any alternative response. The steered alternative provides a meaningfully better reference point for judges than either random resampling or prompt-based approaches. We note that honesty prompting shows promise as an alternative approach, though our focus here is on validating steering-based evaluation.
>
> > **W2: "Limited models: Agent is only Gemma-2B-it"**
>
> We have added experiments with Llama-3.1-8B-Instruct (Section 6.3.3, full results in Appendix B). We find very similar patterns: Single (0.832) → Paired (0.903) → JUSSA (0.892). While JUSSA shows slightly smaller gains on Llama, we note that no extensive hyperparameter optimization was performed for Llama due to computational constraints, so we expect better steering vector configurations could improve these results further.
>
> > **W3: "Residual stylistic cues: […] gains may still be partly driven by surface cues (length, boilerplate)"**
>
> We investigated this through multiple analyses in Appendix A:
>
> Our embedding similarity analysis (Appendix A.4) computed Mean Embedding Cosine Similarity (MECS) and Generation Entropy (GE) metrics across all response types. For the manipulation dataset, MECS values remain consistent (0.954-0.962) and GE shows minimal variation (7.04-7.10), indicating that steering does not create obvious surface-level patterns or reduce linguistic fluency.
>
> Based on your suggestions we also analyzed response lengths and added these to Table 5. **For the manipulation dataset, mean lengths are very similar across response types (base: 947±90, provoked: 915±199, base-steered: 950±98, provoked-steered: 976±76), with differences well within standard deviations**. For the sycophancy dataset, steering does reduce response length for both base (593→350) and provoked (679→385). However, this symmetric reduction in both conditions means length differences cannot explain JUSSA's improved discrimination between base and provoked responses, since AUROC measures the relative difference between these two conditions rather than absolute scores.
>
> In our human annotation study (Section 5 and Appendix A.1), annotators rated manipulativeness after seeing randomized response orders. If gains were purely stylistic, human judgments would likely diverge from LLM judges. Instead, both agreed that steered responses were less manipulative, suggesting the effect is content-based.
>
> We also manually inspected specific phrase patterns (Appendix A.2). For example, "I am an AI" appears in only 25% of steered responses despite increased honesty, showing steering doesn't simply default to boilerplate disclaimers.
>
> These analyses suggest JUSSA's gains are primarily driven by content changes rather than superficial cues.
>
> > **W4: "Dataset external validity: […] largely model-authored"**
>
> We acknowledge this is a valid limitation. Our dataset is model-generated, but underwent extensive human validation: 130 questions across 13 categories with manual annotation (Appendix A.1) and detailed manual inspection (Appendix A.2). The agreement between human annotators and LLM judges on manipulativeness rankings provides evidence that the patterns generalize beyond model-specific artifacts. That said, we agree that incorporating human-written prompts in future work would strengthen the findings.

---

### Author Response · Authors · 2025-11-24
**General Response**

We thank all reviewers for the time and effort spent reviewing our paper.

We are pleased that all reviewers recognize the novelty of our work, the significance of our experimental results, and the relevance of our contributions.

We have revised our paper based on the feedback and made the following modifications.
1) **Lack of baselines (Reviewers nxs6, 85f2, To6N)**: As it was unclear whether the effect of JUSSA came from the steered alternative or any alternative, we have included baseline results in the new Section 6.3.1. We compare our model with a resampled alternative using a higher temperature (1.0) and with an alternative which is prompted for honesty. We found that steering vectors achieve the highest AUROC (0.911), while honesty prompting (0.876) provides modest gains and resampling (0.778) degrades performance below the single-response baseline (0.846).
2) **Limited models tested (All Reviewers)**: We have now run experiments with Llama-3.1-8B-it and agree that this provides a better indication of generalizability. Full results are in the updated paper, Section 6.3.3 and more in Appendix B. In short, we find very similar patterns to Gemma-2-2b-it, with global AUROC scores of 0.832 (Single), 0.903 (Paired), and 0.892 (JUSSA).
3) **Ablation of steering vector's sample (Reviewer 8vA5)**: We have rerun experiments using a sample from each of the 9 verified categories to assess stability, see Section 6.3.2. We find small variance across samples, with JUSSA improving upon the Single Judge setup in 8/9 categories.
4) **Missing Related Work (Reviewers 8vA5 and To6N)**. We have modified the Related Work section to now position our work better among other works related to the LLM-as-a-judge.
5) **Minor correction**: The paper stated that AUROC scores would exclude the 4 noisy manipulation categories from Section 6 onward, but the results from Figure 5a included them. This has been corrected, and all updated AUROC scores reflect only the 9 verified categories. Note that the plot is almost exactly the same except with increased scores for all.

We believe we have addressed the main concerns raised by the reviewers and are grateful for their suggestions, which have strengthened our findings.

---

### Meta-Review · Area_Chair_L2Ny · 2025-12-09

**Summary:**

The paper reframes activation steering as an evaluation aid, using a lightweight single-sample vector to generate honest counterfactuals that help judges detect subtle manipulation and sycophancy. It offers layer-level insight, shows consistent AUROC gains in several settings, and includes human validation for the manipulation dataset. However, baselines are incomplete and may confound the effect of steering, and results rely on one small agent model and largely model-authored data. Prompt framing and residual stylistic cues could bias judges, and statistical rigor and cross-model generalization are limited. Broader benchmarks and comparisons to recent judge-improvement methods are needed to establish external validity.

**Reviewer Concerns:**

Addressed: Added baselines disentangle the effect of steering vs any alternative, broader model coverage includes Llama-3.1-8B-it with similar patterns, steering-sample ablations show low variance across categories, related work positioning is strengthened, and the AUROC reporting bug is corrected to use only the 9 verified categories.

Outstanding: External validity would benefit from larger agents and additional families, broader third-party honesty and manipulation benchmarks, stronger statistical rigor (confidence intervals, multi-seed), and tighter controls for prompt framing or stylistic cues that might bias judges.

**Reviewer Scores:**

No discussion incurred, so it is hard to confirm.

---

### Decision · Program_Chairs · 2026-01-26

Reject